# Hemophagocytic Lymphohistiocytosis Secondary to Hodgkin’s Lymphoma with Isolated Bone Marrow Involvement in a Newly Diagnosed HIV Patient

**DOI:** 10.3390/medicina59071274

**Published:** 2023-07-09

**Authors:** Alberto Lopez-Garcia, Laura Solan, Beatriz Alvarez, Juan Carlos Caballero, Javier Cornago, Laura Pardo, Francisco Javier Diaz de la Pinta, Raul Cordoba, Maria Rodriguez-Pinilla

**Affiliations:** 1Department of Hematology, Fundación Jiménez Díaz University Hospital, 28040 Madrid, Spain; laura.solan@quironsalud.es (L.S.); juan.caballero@quironsalud.es (J.C.C.); javier.cornago@quironsalud.es (J.C.); laura.pardo@quironsalud.es (L.P.); raul.cordoba@fjd.es (R.C.); 2Instituto de Investigacion Sanitaria, Fundación Jiménez Díaz University Hospital, 28040 Madrid, Spain; smrodriguez@fjd.es; 3Department of Infectious Diseases, Fundación Jiménez Díaz University Hospital, 28040 Madrid, Spain; balvarez@quironsalud.es; 4Department of Pathology, Fundación Jiménez Díaz University Hospital, 28040 Madrid, Spain; fjavier.diazp@quironsalud.es

**Keywords:** Hodgkin’s lymphoma, human immunodeficiency virus, Epstein–Barr virus, hemophagocytic lymphohistiocytosis

## Abstract

Human immunodeficiency virus (HIV) infection is known to be associated with the development of Hodgkin’s lymphoma (HL). Exclusive extranodal bone marrow involvement is less common. Co-infection by other viruses, such as the Epstein–Barr virus (EBV), increases the incidence of a frequent complication denominated by hemophagocytic lymphohistocytosis (HLH). We present the case of a 50-year-old patient with the above clinical spectrum who develops several serious complications during treatment.

## 1. Introduction

Hodgkin’s lymphoma (HL) is a lymphoproliferative syndrome frequently associated with patients with human immunodeficiency virus (HIV) infection, although it is not currently recognized as a defining entity of human immunodeficiency syndrome (AIDS) [1]. Extranodal involvement is common in this situation, although it is usually accompanied by lymph node involvement [2,3]. Co-infection with the Epstein–Barr virus (EBV) may play a role in the etiology of this type of lymphoma in this population, but the relationship is not well established [4]. The combination of the three previous entities has been defined as causing the development of hemophagocytic lymphohistiocytosis (HLH), requiring specific management. We present a clinical case presenting Hodgkin’s lymphoma as an initial manifestation of HIV infection and EBV coinfection associated with HLH and severe complications during treatment [5].

## 2. Case Presentation 

A 50-year-old Uruguayan man with no personal history of interest presented to the emergency department with a four-month history of asthenia, anorexia, profuse nocturnal diaphoresis, weight loss of up to 10 kg, and intermittent fever with associated dystrophic sensation. The patient reported risky sexual relations, with the last serology screening for HIV infection three years earlier being negative. 

Relevant clinical findings on admission included a striking masseteric and temporal atrophy, oral thrush, bilateral laterocervical subcentimeteric lymphadenopathy, predominantly on the left side, residual lesions of metameric herpes zoster on the left side, and a palpable hepatosplenomegaly. In addition, the patient had a body temperature of 39 °C. 

The hemogram showed pancytopenia (leukocytes (Leu) 3.16 × 10^3^/µL (0.9 × 10^3^/µL lymphocytes and 1.9 × 10^3^/µL neutrophils), hemoglobin (Hb) 10.8 g/dl, platelets (Pts) 134,000 × 10^3^ µL, as well as an ultrasensitive C-reactive protein of 19 mg/dl. 

A full-body computed tomography (CT) scan showed a hepatomegaly of approximately 23 centimeters (cm), without focal lesions, as well as lymphadenopathies in the upper abdomen, hepatic hilum up to 29 millimeters (mm), and retroperitoneal and mesenteric up to 11 mm in subcentimeter iliac and inguinal regions.

Flow cytometry was requested due to alterations in the hemogram showing severe lymphopenia at the expense of all natural killer, B, and T lymphocyte subpopulations, the last one presenting a significant decrease in the TCD4+ cell count (<100 cells/µL) with a preserved TCD8+ cell count. In addition, circulating lymphoplasmoid cells were identified without light chain restriction, suggesting normality. 

Based on these findings, blood and urine cultures were requested and were normal. A serology for multiple viruses showed positive HIV1 and HIV2 antibodies and p24 antigens. The confirmatory Western blot test showed positive antibodies against HIV1 gp160, gp41, and p24 proteins. Regarding viral load, quantification of HIV-1 viral RNA was 72,100 copies/mL (4.86 log) (undetectable values < 20). Given the HIV infection, determination of lymphocyte populations was performed. The TCD4+ lymphocytes in absolute value resulted in 70 cells/microliter with a CD4/CD8 ratio of 0.14. Resistance mutation studies for HIV antivirals were also performed without finding any resistance to them.

In addition, anti-core antibodies for hepatitis B virus were positive; the viral load was measured by DNA and the rest of the study was negative, presenting the patient as a carrier of this virus within the study. The rest of the study showed no relevant alterations with negativity for hepatitis C, syphilis, and tuberculosis. Leishmania serology was also performed due to the patient’s pancytopenia, with negative results.

At that time, the patient started highly active antiretroviral therapy (HAART) (Bictegravir 50 mg, Emtricitabine 200 mg, and Tenofovir alafenamide 25 mg per day) and empiric antibiotherapy, started at emergency department, was suspended due to the low clinical suspicion of bacterial infection. After HAART initiation, HIV viral copies decreased rapidly to 67 copies/mL (1.83 log) three months later and undetectable values after six months of treatment. CD4 lymphocytes reached levels above 100 cells/microliter eight months after HAART.

Bone marrow (BM) aspirate was performed and showed no cytological data or dysplasia, infiltration by a lymphoproliferative syndrome, or plasma cell dyscrasia. Marrow blood immunophenotyping was normal, as were the marrow cultures. 

BM biopsy revealed isolated ample cytoplasmic cells with one or more nuclei with prominent nucleoli-expressing CD15, CD30, PAX5, and EBER as well as images of hemophagocytosis with hardly any residual normal hematopoietic tissue. Reed–Sternberg cells were observed. In addition, large numbers of polytypic plasma cells and a striking increase in CD3/CD8-positive small T lymphocytes and abundant loose epithelioid histiocytes were also shown, with no granuloma formation. These cells did not show expression for CD20 or CD79a. Furthermore, no human herpes virus 8 (HHV8) positive cells or an increase in the number of blasts were observed (Figure 1).

These findings make the diagnosis of Hodgkin’s lymphoma (HL) more likely. 

In order to confirm and stage the disease, a positron emission tomography CT (PET-CT) scan was requested, revealing multiple supradiafragmatic and infradiafragmatic adenopathies, most of them with discrete–moderate fluorodeoxyglucose (FDG) uptake (some with intense activity). The one at the right inguinal region was the most accessible for biopsy (Figure 2).

In addition, FDG uptake was revealed in the gastric fundus, showing only data of gastritis, hepatomegaly without significant alterations to the metabolism, and a discrete diffuse hypermetabolism in BM. No alterations were found in the rest of the study. 

A lymph node biopsy was performed showing hyperplasia of germinal centers with increased vascularization in the centers and interfollicular area with expansion and fibrosis of the capsule. Monocytic cells were revealed in subcapsular sinusoids, histiocyte clusters, and numerous plasma cells in the centers and perifollicular area that did not show light chain restriction. Abundant CD30 cells were revealed in the centers and interfollicular area with numerous EBV (EBER)-positive cells in the centers, although in smaller numbers, surrounding them. These findings were compatible with the histologic diagnosis of HIV-related lymphoid hyperplasia associated with possible EBV reactivation (Figure 3).

As part of the etiological study, the viral load of Epstein–Barr virus measured by the DNA polymerase chain reaction (PCR) in peripheral blood using probes that hybridize in the EBNA-1 protein gene was requested and resulted in 47,406 copies/mL. 

With these findings, the striking clinical improvement after initiation of antiretroviral therapy and the rarity of Hodgkin’s disease isolated to marrow made us take the results of the first marrow biopsy with caution, delaying systemic treatment with chemotherapy.

Later, four weeks after initiation of HAART, the patient’s clinical condition worsened, making HL the likely diagnosis. 

Subsequently, the patient developed more pronounced pancytopenia (Leu 0.61 × 10^3^/µL, Hb 6.6 g/dL, Pts 42,000/µL), recurrence of fever, persistent splenomegaly on physical examination, hyperferritinemia (8190 ng/mL, (elevated CD25 soluble interleukin receptor (>7500 U/mL), and hypertriglycidemia (275 mg/dL), which is associated with the phenomena of hemophagocytosis observed in the histological study and suggested HLH.

Regarding the underlying cause of HLH, HIV infection, EVB infection, Hodgkin’s lymphoma, or a combination of several of the above could trigger the current clinical situation. According to all these data, HL limited to bone marrow, HIV and EBV infection and the development of HLH secondary to its lymphoproliferative syndrome, the case was discussed by the multidisciplinary committee on lymphoproliferative neoplasms. Treatment was initiated according to the Histiocyte Society protocol (HLH-94), which included descending doses of dexamethasone and etoposide. Lumbar puncture was performed showing no disease, so triple intrathecal therapy was excluded from the treatment.

In order to control EBV infection as a possible underlying trigger of HLH, monoclonal antibody Rituximab was added to the treatment. After starting Rituximab on 15 March 2022, the patient’s EBV viremia was negative and remains negative to date (Figure 4).

Finally, for the treatment of HLH, the ABVD regimen (anthracycline, bleomycin, etoposide, dacarbazine) was decided upon for six cycles.

During treatment, the patient presented multiple clinical complications, mainly hematologic toxicity and cytomegalovirus infection, which caused esophageal ulcers (Figure 5) leading to a stenosis that required enteral nutrition.

Currently, the patient is being monitored and remains in complete remission with all secondary complications resolved with a current follow-up of 27 months. 

## 3. Discussion

As previously described, we present the case of a patient with primary HIV and EBV co-infection. One of the elements that makes this a particular case is the exclusive involvement of the bone marrow by an HL. The secondary development of HLH made this a challenging and complex case to manage. Despite a broad differential diagnosis for esophageal stenosis being proposed, it finally turned out to be CMV involvement. Although the response rate for the treatment of HLH is variable, in many cases it causes long-lasting sequelae or high mortality [6]. In our case, treatment for HL and HLH was successful, and the patient is alive and disease-free after 27 months of follow-up.

Regarding the initial presentation, the clinical findings, pancytopenia, and both analytical and CT imaging findings made it essential to perform a BM workup to reach a definitive diagnosis. 

In patients with HL and HIV infection, extensive extranodal involvement is frequent, including BM infiltration, although exclusive involvement of the BM is infrequent. This fact is probably explained by the underdiagnosis of this entity [1].

Indeed, the fact that HL is not an AIDS-defining disease is an issue that will probably have to be reviewed in the future, since its incidence in immunocompromised populations, such as HIV patients, is high [1].

On another note, as part of the diagnosis, a lymph node biopsy was performed which, although the incidence of lymph node involvement in this population is high, the histological findings were compatible with related lymphoid hyperplasia.

The reasons for this conclusion were that these findings in an inguinal lymph node biopsy can occur in diseases other than HL, such as peripheral B- and T-cell lymphomas, infectious mononucleosis EBV, and follicular hyperplasia [7].

Importantly, the prevalence of EBV infection globally is estimated to be around 90%, contributing to Hodgkin’s/Reed–Sternberg cell survival (HRS) [8].

This circumstance, added to the high viral load of EBV demonstrated by PCR in the peripheral blood, made us consider related lymphoid hyperplasia as a diagnosis and not another cause contemplated in the differential diagnosis [3,8,9].

Finally, with respect to diagnostic features, the diagnosis of HLH is by exclusion. It is a life-threatening entity that must be considered in patients with cytopenias, increased transaminases, triglycerides and ferritin, coagulopathy, elevated IL-2 receptor, and decreased or absent NK cells, as well as the appearance of fever and splenomegaly [6]. The presence of hemogagocytosis in lymph nodes or marrow alone is not diagnostic of this pathology. HLH is a potentially fatal condition. It can present in a primary or pediatric form caused by genetic defects, or in a secondary form. Both have in common an uncontrolled activation of TCD8+ cytotoxic cells as the underlying abnormality in most forms of HLH [5].

Regarding the secondary causes that can trigger this condition, autoimmune diseases, infections, malignant diseases, and drugs, among others, may be involved [10].

In the presented patient’s clinical situation, HIV infection, EVB infection, HL, or a combination of the above could be causing the current clinical status.

Although the criteria for HLH may be debated, the clinical presentation was compatible with this pathology, which contributed to the decision to start the HLH-94 protocol [11] with descending doses of dexamethasone and etoposide. This protocol is derived from the primary or pediatric HLH. Five-year survivals according to collaborative studies can reach 62%, but early mortality and neurological sequelae might be as high as 20% [12]. As in other types of lymphoma, CNS involvement makes it necessary to manage these patients with triple intrathecal chemotherapy, which was not necessary in our case. A new protocol for this same society is being carried out (HLH-2004) which includes cyclophosphamide as induction therapy, although to date we still do not have solid data [5].

As in some other cases described in the literature, the coincidence of HL in the bone marrow together with the development of HLH made this case difficult to manage and diagnose, since the symptoms may overlap with each other, making interpretation difficult [13,14].

Regarding therapy and referring to HL, the treatment of advanced HL, defined as the accumulation of different criteria of a poor prognosis such as age over 45 years, stage IV, decreased serum albumin levels, male sex, anemia, and lymphocytosis or lymphopenia, is based on intensive BEACCOP type treatments (bleomycin, etoposide, adriamycin, cyclophosphamide, vincristine, procarbazine, and prednisone). This is the recommended regimen for fit patients [15].

The rationale for choosing a less intensive ABVD-type regimen was based on the decision of the multidisciplinary lymphoma committee not to increase the added toxicity to the regimen for the treatment of HLH [16].

In addition, in patients with HL associated with lymphoproliferative neoplasms, initiation of HAART therapy is indicated [17].

Despite HAART, incidence of HL has not decreased since its implementation and this could be explained in part by the dependence of HRS on CD4+ T lymphocytes [18].

Chronic antigenic stimulation and co-infection with other oncogenic viruses appear to be key factors for lymphomagenesis in HIV patients. This may be the reason why even in the HAART era the incidence of neoplasms is higher in patients living with HIV than in the general population [19].

Finally, in order to control EBV infection as a possible underlying trigger of HLH, monoclonal antibody Rituximab was added to the treatment as it is thought that it can eliminate EBV-infected B cells [20].

As a consequence of all the treatments administered, the patient presented retrosternal pain and dysphagia. A gastroscopy was performed, and the biopsy was compatible with CMV infection. In addition, the patient developed cytopenia that required transfusion support and the delay in the treatment to be resolved. Despite the above, with a follow-up of almost two and a half years, the patient remains alive and in complete remission with symptoms derived from CMV esophagitis. Close ambulatory observation remains.

## 4. Conclusions

The incidence of HL in the HIV population is increasing. Its presentation is mainly advanced, with exclusive medullary involvement being less frequent. EBV co-infection may play an important role in the etiopathogenesis of this type of lymphoma, with an increased incidence even in the HAART era. The three previous entities can produce HLH, a potentially fatal entity that requires specific treatment for its management. Multidisciplinary patient cooperation is fundamental for the treatment of these entities.

## Figures and Tables

**Figure 1 medicina-59-01274-f001:**
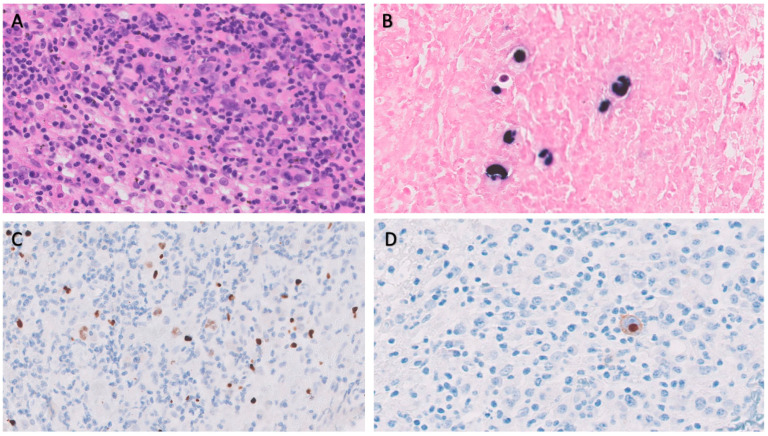
Bone marrow biopsy. (**A**): Hematoxylin eosin stain × 40. (**B**): EBER in situ hybridization. (**C**): PAX5. (**D**): CD30.

**Figure 2 medicina-59-01274-f002:**
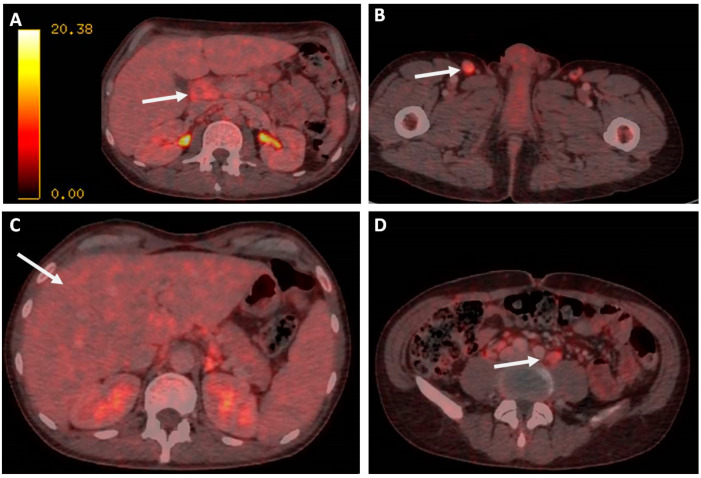
PET-CT scan images. (**A**): The hepatic hilum lymphadenopathy is marked with arrows. (**B**): Inguinal lymphadenopathy. (**C**): Hepatomegaly. (**D**) Mesenteric lymphadenopathy.

**Figure 3 medicina-59-01274-f003:**
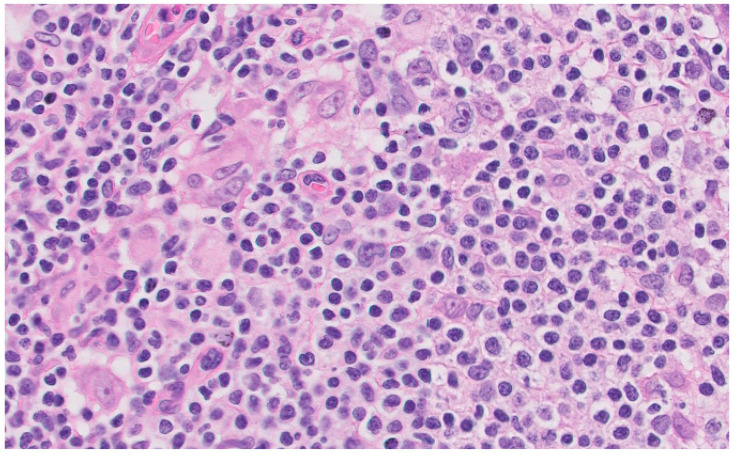
Inguinal lymph node biopsy. Hematoxylin eosin stain × 40.

**Figure 4 medicina-59-01274-f004:**
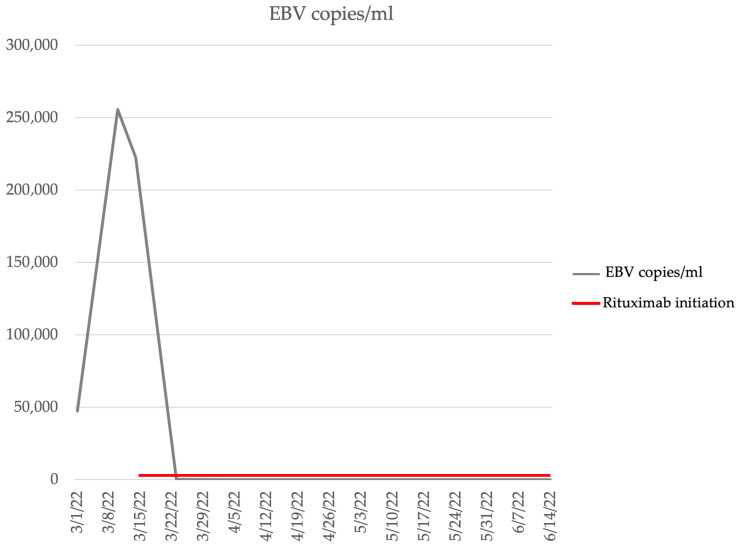
EBV viral copies after Rituximab initiation.

**Figure 5 medicina-59-01274-f005:**
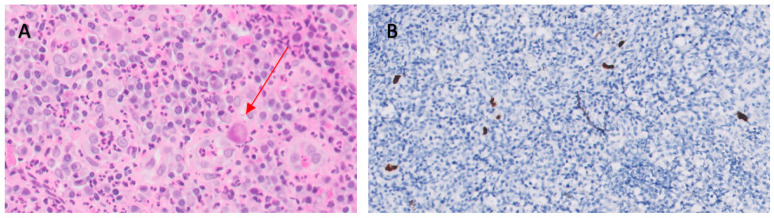
CMV ulcers. (**A**): Hematoxylin eosin stain × 40. Arrow-headed cell with owl’s eye appearance typical of CMV infection. (**B**): CMV.

## Data Availability

The data presented in this study are available on request from the corresponding author.

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
