# Peer review of "Hemophagocytic Lymphohistiocytosis Secondary to Hodgkin’s Lymphoma with Isolated Bone Marrow Involvement in a Newly Diagnosed HIV Patient"

_medicina, 2023, doi:10.3390/medicina59071274_

Round 1

Reviewer 1 Report

-Is the diagnosis of Hodgkins certain I could not pic that from the manuscript if not the title should change to lymphoproliferative disease or to send the manuscript to a pathologist to revise.

-What was the H-score I beleive this should be included for adults with secondary HLH H-score is more reliable than HLH-2004

-The author should mention the HIV status upon initiating rituximab and the CD4 counts as if itis less than 50 will be associated with more risk of infection.

-The author should mention the interium PET after 2 cycles of ABVD was he PET -ve

-Page 4 the last paragraph (finally,for treatment of HLH the ABVD,.....)

I beleive he mean the treatment of hodgkins not HLH

Author Response

Dear Editors

Please find attached the proposed changes, after suggesting a minimum of 2500 words and checking that the references are relevant to the content of the manuscript.

Thank you very much 

Reviewer 2 Report

Nice case report!

The authors tried to present an interesting case of hemophagocytic lymphohistiocytosis secondary to Hodgkin Lymphoma in a simple and understandable way.

The manuscript requires minimal changes, primarily changes of a technical nature (see pdf with comments).

In the Discussion, literature data related to the same topic that is covered in this paper should be cited, and not repeat items that have already been presented in the Results section!

Author Response

Response to Reviewer 2 Comments

The authors tried to present an interesting case of hemophagocytic lymphohistiocytosis secondary to Hodgkin Lymphoma in a simple and understandable way.

Point 1: The manuscript requires minimal changes, primarily changes of a technical nature (see pdf with comments).

Response 1: Thank you very much for your kind comments that make our article better with each review. We have made a thorough revision thanks to your suggestions. We have modified all the technical issues you are referring to.

Point 2: In the Discussion, literature data related to the same topic that is covered in this paper should be cited, and not repeat items that have already been presented in the Results section!

Response 2: Once again we reiterate our thanks for your comments.  We have added some important citations to the manuscript related to other articles that are related to ours.

In addition, we have modified the discussion to avoid some aspects that could seem redundant.

Reviewer 3 Report

Alberto Lopez-Garzia et al present a very interesting case report. This is highly relevant for the readers of the highly esteemed journal Medicina and for colleagues worldwide. 

However, at the moment the infectious aspects are far too superficial, to say the least. This concerns both the "Case presentation"  and the "Discussion" section. 

Furthermore, an additional illustration with the CT scans would be very useful. 

For the reasons mentioned above, from my point of view major revisions are urgently needed. I encourage the authors to do this as soon as possible. I am already very much looking forward to a revised version. 

Minor editing of English language required

Author Response

Response to Reviewer 3 Comments

Alberto Lopez-Garzia et al present a very interesting case report. This is highly relevant for the readers of the highly esteemed journal Medicina and for colleagues worldwide. 

Point 1: However, at the moment the infectious aspects are far too superficial, to say the least. This concerns both the "Case presentation"  and the "Discussion" section. 

Response 1: Thank you very much for your kind comments that make our article better with each review. We have carried out a thorough and systematic review of the manuscript with all the aspects suggested to us.

We have added relevant information regarding the infectious approach, adding some infections that were ruled out, relevant information related to HIV infection and a graphic showing the effectiveness of RItuximab treatment for Epstein-Barr Virus viremia.

We hope you appreciate it.

Point 2: Furthermore, an additional illustration with the CT scans would be very useful. 

Response 2: As you have mentioned, a figure with an X-ray examination of the patient is very useful to illustrate the case. We have added a PET-CT scan image of the patient, which we consider provides even more information than the conventional CT scan. Thank you for the contribution.

Point 3: For the reasons mentioned above, from my point of view major revisions are urgently needed. I encourage the authors to do this as soon as possible. I am already very much looking forward to a revised version. 

Response 3: As mentioned above, we have carried out a thorough revision of the manuscript, adding information relevant to the case and eliminating some information that might be somewhat redundant. We hope you like the modifications.

Round 2

Reviewer 3 Report

Due to the comprehensive revision, the quality of the manuscript has developed excellently.

From my point of view, it should be published as it is as soon as possible, because it has a very high scientific and clinical value.

I congratulate the authors on this now wonderful scientific work. It is now eminently suitable for the highly esteemed journal Medicina.